# Effects of Soy Protein Concentrate in Starter Phase Diet on Growth Performance, Blood Biochemical Indices, Carcass Traits, Immune Organ Indices and Meat Quality of Broilers

**DOI:** 10.3390/ani11020281

**Published:** 2021-01-22

**Authors:** Qianyun Zhang, Shan Zhang, Guanglei Cong, Yijian Zhang, Marianne Hjøllund Madsen, Benjie Tan, Shourong Shi

**Affiliations:** 1College of Animal Science and Technology, Guangxi University, Nanning 530004, China; 17858765721@163.com (Q.Z.); ZhangYiJian1105@163.com (Y.Z.); 2Poultry Institute, Chinese Academy of Agriculture Science, Yangzhou 225125, China; zhangshan3321@163.com (S.Z.); MZ120181027@yzu.edu.cn (G.C.); 3Jiangsu Co-Innovation Center for the Prevention and Control of Important Animal Infectious Disease and Zoonoses, Yangzhou 225000, China; 4TripleA a/s, Bjoernkaervej 16, DK-8783 Hornsyld, Denmark; mhm@triple-a.dk

**Keywords:** soy protein concentrate, growth performance, carcass traits, meat quality, immune organ indices, blood biochemical indices, broiler

## Abstract

**Simple Summary:**

It is well known that early nutrition plays an important role in determining changes in later physiology. Therefore, in order to promote the early growth of broilers, it is necessary to feed diets with high concentration of nutrients and high digestibility in the early stage after hatching. Since soy protein concentrate (SPC) has the characteristics of low content of antinutritional factors (ANFs), high digestibility and high amino acid content, we evaluated whether supplementation of SPC in starter diet could promote the later growth of broilers without changing the diets nutritional level (the same Metabolizable energy (ME), crude protein (CP) and amino acids (AAs)). The results showed that supplementing SPC in starter diet could improve the growth performance, carcass traits and immune organ indices of broilers. The results are very promising and SPC can be considered as a starter diet additive for commercial use. However, the mechanism of adding SPC in starter diets to improve broiler performance needs further study.

**Abstract:**

Soybean meal (SBM) is high in antinutritional factors (ANFs), which is not conducive to the starter growth of broilers. The purpose of this study was to investigate the effects of soy protein concentrate (SPC) in starter diet on growth performance, carcass traits, meat quality, immune organ indices and blood biochemical indices of broilers. A total of 384 1-day-old Arbor Acres (AA) male broilers (46.05 ± 0.37 g) with similar body weight were randomly divided into 4 groups with 8 replicates in each group and 12 broilers in each replicate. The experiment was divided into three phases: in starter phase (1–10 d), birds were fed a corn-SBM-based basal mash diet (control) and the basal diet was supplemented with SPC at 4% (SPC4), 8% (SPC8), 12% (SPC12). In the grower phase (11–21 d) and the finisher phase (22–42 d), the birds in all four treatment groups were fed the same diets. The results showed that the body weight was significantly increased in the SPC8 and SPC12 groups of broilers at 10 d and 42 d (*p <* 0.05). The average daily gain was significantly increased in the SPC12 group of broilers at 1–10 d and 1–42 d (*p* < 0.05). The average daily feed intake was significantly increased in the SPC8 and SPC12 groups of broilers at 1-10 d (*p* < 0.05). The feed conversion rates at 1–42 d (*p* = 0.055) tended to decline in the SPC12 group. The carcass yield and the thymus indices were significantly increased in the SPC12 group of broilers at 42 d (*p* < 0.05). Alanine aminotransferase (ALT)/aspartate aminotransferase (AST) tended to decline in SPC12 group at 10 d (*p* = 0.055) and total protein (TP) tended to increase in the SPC12 group at 42 d (*p* = 0.080). The contents of total cholesterol (T-CHO) and high-density lipoprotein (HDL) were significantly elevated in the SPC12 group of broilers at 42 d (*p* < 0.05). In conclusion, dietary inclusion of 12% SPC as a starter diet can be recommended due to the positive effects on broilers.

## 1. Introduction

Epidemiological studies in humans and research on animal models during the last two decades have provided strong evidence that the early environment, including early nutrition, plays an important role in determining changes in later physiology [1]. With the continuous shortening of broiler growth time, the starter growth of broilers becomes more and more important. Researchers also found that the starter growth performance of broilers affects the performance of the broilers throughout all growth phases [2]. At the starter stage after hatching, the development of the digestive tract is imperfect and the digestion and absorption ability are not strong [3]. Consequently, to promote the starter growth of broilers, the diet must have a high concentration of nutrients and high digestibility [4].

Soybean meal (SBM) is a common protein source for poultry feed and using SBM as feed material for poultry can promote good production performance [5], due to its high nutritional value and balanced amino acid composition. However, SBM contains antinutritional factors, such as protease inhibitor, non-starch polysaccharide, phytate and lectin [6]. These antinutritional factors will damage the structure of the animal digestive tract, reduce the utilization rate of proteins and minerals, inhibit animal feeding, disrupt the balance of the intestinal microecology and have adverse effects on animal growth [7,8]. Therefore, there is a need for a feasible method to reduce the antinutritional factors in order to ensure the intestinal development of young animals. Soy protein concentrate (SPC) is a kind of soybean protein product with low antinutritional factor, high digestibility and high amino acid level [9]. The processes involved in SPC preparation may result in improved energy and amino acid availability, which, in turn, may have some positive effects on chick growth, especially during the first 21 d post hatch, because the energy and amino acid digestibility would increase with age [10,11,12]. However, there are few studies on the application of SPC in the starter growth stage of broilers. On the one hand, previous studies mainly focused on adding SPC in the whole feeding stage of broilers, which led to the increase of feed cost [12]. On the other hand, the low-cost antibiotics were allowed to be added to diets to promote early growth of animals [13,14]. But in recent years, with the prohibition of antibiotics, the importance of early diet has attracted the attention of researchers.

The aim of the study was added SPC to the starter diets of broilers to determine its effect on growth performance, carcass traits, meat quality, immune organ indices and the blood biochemical indices of broilers, in order to provide evidence for the practical production of broilers. 

## 2. Materials and Methods

### 2.1. Soy Protein Concentrate

AX3Digest^®^ (produced by TripleA company in Denmark, Hornsyld, Denmark) is a soy protein concentrate produced by a novel patented method. AX3Digest^®^ has high digestibility and a minimal content of antinutritional factors due to the special production process. This study compared the nutritional quality in SBM and soybean concentrate protein (Table 1).

### 2.2. Broilers, Experimental Design and Experimental Diet

This study was performed at the Institute of Poultry, Chinese Academy of Agricultural Sciences, Yangzhou, China (32°31′17.56″ N, 119°30′30.97″ E, 6.4 m above sea level), from November 2019 to January 2020. Experimental procedures were performed after receiving approval from the Local Experimental Animal Care Committee and the ethical institutional committee of the Institute of Poultry, Chinese Academy of Agricultural Sciences.

A total of 384 one-day-old Arbor Acres (AA) male broilers (46.05 ± 0.37 g) were purchased from Jiangsu Jinghai Poultry Co., Ltd., Haimen, China. The birds were randomly distributed into 4 treatment groups with 8 replicates for each treatment and 12 birds in each replicate in a completely randomized design. The birds were housed in wire-floored pens (1.2 m × 0.8 m × 0.6 m) in an environmentally controlled room at a kept temperature of 33 degrees Celsius for the first three days. The environmental temperature was then gradually decreased by 1 degrees Celsius every two days until the final temperature reaches 20 degrees Celsius. The lighting schedule was continuous light for 3 days after hatch, followed by a schedule of 23 h of light and one hour of darkness throughout the test. Meanwhile, all broilers had ad libitum access to feed and fresh water. All birds were vaccinated against Bronchin on days 7 and 21 and against Gumboro on days 14 and 28. 

During the whole rearing period, all broilers were randomly divided into four dietary treatments and the experimental diets were fed in three phases: the starter phase (days 1 to 10), the grower phase (days 11 to 21) and the finisher phase (days 22 to 42).In starter phase (1–10 d), birds were fed a corn-SBM-based basal mash diet (control) and the basal diet was supplemented with SPC at 4% (SPC4), 8% (SPC8), 12% (SPC12). In the grower phase (11–21 d) and the finisher phase (22–42 d), the birds in all four treatment groups were fed the same diets. All experimental diets had the same nutritional level (the same ME, CP and AAs) The formulation and composition of the commercial broiler chick basal diet was calculated according to NRC (1994) guidelines (Table 2).

### 2.3. Growth Performance

The broiler chicks were individually weighed on day 1, day 11, day 22 and day 42. Mortality was observed and recorded. The performance variables measured in this study include body weight (BW), average daily gain (ADG), average daily feed intake (ADFI) and feed conversion rate (FCR). 

### 2.4. Serum Biochemistry Indices

After cervical dislocation, blood samples (3 mL) were collected from the jugular vein of eight birds in each group, at the following time points: 10 and 42 days of life. The blood was centrifuged at 4 °C for 10 min at 4000 rpm to take the supernatant to obtain a serum sample and stored at −20 °C for serum biochemical indicators. A chemistry analyzer (C-8000, Abbott, Chicago, IL, USA) was used for the determination of the following biochemical parameters of the serum: aspartate transaminase (AST, U/L), alanine transaminase (ALT, U/L), total protein (TP, g/L), globulin (GLO, g/L). urea nitrogen (BUN, mmol/L), total cholesterol (T-CHO, mmol/L), triglyceride (TG, mmol/L), high-density lipoprotein (HDL, mmol/L) and low-density lipoprotein (LDL, mmol/L).

### 2.5. Carcass Traits

At the end of the 42-day experiment, 32 broilers (8 in each group) were sampled randomly for carcass evaluations and were weighed and manually slaughtered. The weights of the carcasses, liver, gizzard, heart, thigh, breast and abdominal fat were recorded and expressed as g/kg of live body weight, as follows:Carcass yield (%) = (carcass weight (g)/live body weight (g)) × 100,(1)
Organ index (g/kg) = organ weight (g)/live body weight (kg),(2)

### 2.6. Immune Organ Indices

On the day 10 and day 42, the birds were exsanguinated to obtain tissues. The spleen, thymus (on the day 10, the thymus is too small to weight, so there only index of spleen and bursa of fabricius in day 10.) and bursa of fabricius were collected, rinsed with physiological saline solution, then weights were recorded after blotting with filter papers. The thymus, spleen and bursa fabricius indices were calculated as the relative organ to body weight of the broilers before slaughter.
Immune organ index (g/kg) = Immune organ weight (g)/live body weight (kg).(3)

### 2.7. Meat Quality

Pressing loss measurements were performed according to the method of Farouk et al. (2004) [15]. A breast muscle sample 1.0 cm in diameter and 0.5 cm in thickness was removed from the 32 broilers for each treatment at 24 h postmortem. The samples were then weighed (m1) and wrapped in an 8-layer filter paper. Subsequently, they were pressed using a compression machine (YYW-2, Nanjing Soil Instrument, Nanjing, China) at a force of 35 kg for 5 min and then reweighed (m2). The pressing loss was expressed as follows:Pressing loss (%) = (m1−m2)/m1 × 100%.(4)

The color of the breast muscle was measured at 24 h post-mortem. The measurements were taken from three locations for each sample and average values were given. Lightness (L*), redness (a*) and yellowness (b*) values were determined on one bird from each replicate using a Chroma Meter (CR-400, Konica Minolta, Osaka, Japan).

The pH values of breast muscles were measured on one bird from each pen using a portable pH meter (PH-STAR, Matthaus, Germany) at 24 h post-mortem. The pH probe was inserted at an angle of 45° into the pectoralis major and rinsed with deionized water after each measurement. Each sample was measured three times and the average value was taken as the final result.

The shear force value was evaluated according to the method of Chen (2007) with some modifications [16]. Muscle samples from the right pectoral major muscle were kept at 4 °C for 24 h. After packaging and sealing in a boilable bag, the samples were cooked to an internal temperature of 70 °C in a water bath. Ten minutes was required to reach the end-point temperature. Upon reaching the desired temperature, the muscle samples were removed and cooled to room temperature. A 1.27 cm-diameter core was removed from each sample parallel to the myofiber orientation for shearing perpendicularly to the longitudinal orientation of the myofibers using a Digital Meat Tenderness Meter (C-LM4, Tenovo, Harbin, China). The test speed was 5 mm/s and the maximum force needed to cut the strips was expressed in newtons. For each cooked breast muscle, the strip was sheared in three locations and the average was used for data analysis.

### 2.8. Statistical Analysis

Statistical analyses were carried out with SPSS 21.0 for Windows (SPSS Inc., Chicago, IL, USA). The statistical differences between the treatment groups were determined by a Duncan test. All data are presented as means and combined SEM. Significance (*p*-value) was evaluated at the 0.05 level. For performance data, replicate means used as the test unit for statistical analysis. For data on carcass traits and meat quality, individual birds were considered as the test units.

## 3. Results

### 3.1. Growth Performance

The growth performance of broilers is shown in Table 3. On day 10, compared to the control group, BW and ADG were significantly increased (*p* < 0.05) in the SPC12 group and ADFI was significantly increased (*p* < 0.05) in the SPC8 and SPC12 groups; however, FCR was not affected (*p* > 0.05) by dietary supplementation of SPC. On day 21, compared to the control group, BW, ADG, ADFI and FCR were not affected (*p* > 0.05) by dietary supplementation of SPC. On day 42, compared to the control group, BW was significantly increased (*p* < 0.05) in the SPC8 and SPC12 groups. Furthermore, regarding the overall growth performance of broilers (days 1–42), compared to the control group, ADG was significantly increased (*p* < 0.05) in the SPC8 and SPC12 groups. Meanwhile FCR showed a tendency to decrease (*p* = 0.05) in the SPC12 group. Additionally, compared to the control group, EPI was not affected (*p* > 0.05) by dietary supplementation of SPC.

### 3.2. Serum Biochemical Indices

Data in Table 4 and Table 5 illustrate the impacts of different doses of SPC on the serum biochemical indicators of broilers on day 10 and day 42, respectively. On day 10, compared to the control group, the ALT/AST showed a tendency to decrease (*p* = 0.055) in the SPC12 group. However, other serum biochemical indicators were not affected (*p* > 0.05) by dietary supplementation of SPC. On day 42, compared to the other groups, the content of T-CHO was significantly increased (*p* < 0.05) in the SPC12 group. Meanwhile, compared to the SPC4 and SPC8 groups, the content of HDL was significantly increased (*p* < 0.05) in the SPC12 group.

### 3.3. Carcass Traits

The effect of dietary inclusion of SPC in starter diet on carcass traits is presented in Table 6. Compared to the control group, the carcass yield was significantly improved (*p* < 0.05) in the SPC12 group. However, dietary treatments had no effect on the eviscerated with giblet yield, the yield of thigh, leg, claw, pectoralis major, pectoralis minor, wing, abdominal fat (*p* > 0.05). Meanwhile, there was no effect on the index of heart, liver, pancreas, gizzard and glandular stomach (*p* > 0.05).

### 3.4. Immune Organ Indices

The effects of dietary inclusion of SPC on immune organs are shown in Table 7. On days 10 and 42, the spleen and bursa fabricius index were not affected (*p* > 0.05) by dietary supplementation of SPC. On day 42, compared to the other groups, the thymus index was significantly increased (*p* < 0.05) in the SPC12 group.

### 3.5. Meat Quality

Data on pressing loss, meat color, share force and pH in breast are reported in Table 8. No significant (*p* > 0.05) variations between treatments were observed for all the above physical parameters of meat.

## 4. Discussion

Dietary protein has been one of the nutrients of most interest because the amount and source of protein are often associated with increased susceptibility to diseases such as necrotic enteritis [17]. In terms of protein, feeding highly bioavailable protein sources is possible but not without economic penalty, however, the starter feed intake of broilers only accounts for 3.5% of the feed intake during the entire growth period, which allows more expensive feeds to be added to the starter growth of broilers to regulate their starter growth [3]. In this study, SPC was added to the diet of broilers for 1–10 d to explore its effect on later growth stages. This is in line with the current thinking of starter life nutrition [18]. This developmental pattern is believed to reflect a survival strategy in which great importance is placed on the growth of nutrient supply functions starter in life in order to maximize post-absorptive growth functions later in the life cycle [19,20]. 

Our results showed that adding 8% and 12% SPC in the starter diet of broilers significantly increased the body weight at 10 days and 42 days and significantly increased the daily gain at 1–10 days and 1–42 days. ADFI increased at 1–10 days and 12% SPC significantly reduced the feed to weight ratio at 1–42 days. In parallel with the results of the Jankowski et al. research, replacing SBM with SPC can significantly increase the body weight of a turkey at 8 weeks old [12]. It was also found that adding SPC to the diet of weaned piglets could improve intestinal morphology and thus improve growth performance [21,22]. Jazi et al. [23] reported that replacing SBM with Fermented SBM (FSBM) in the diet can improve growth performance indices of Japanese quail. In line with these results, other studies have also shown that partial inclusion of fermented rapeseed meal [24] and SBM [25] in the diet of broilers can improve weight gain and feed efficiency. These increases may be due to the elimination of most non-starch polysaccharides (NSP) in SPC, which improves the starter nutritional absorption capacity of broilers [26]. In the pre-cecal section of the gastrointestinal tract, soluble NSP may increase the viscosity of the gastrointestinal content, which may disturb the secretion of endogenous enzymes and bile acids, causing morphological changes in the intestine and reducing the digestibility of nutrients [27]. This may result in poorer growth and performance of birds [28]. Therefore, reducing the amount of NSP in diet could be a key means of improving the performance of broilers. However, this conclusion needs to be further verified.

Blood biochemical indices are often used to reflect changes in metabolism and organ function [29]. AST and ALT are the two enzymes with the highest transaminase activity in animals. Under normal conditions, the activities of AST and ALT in the liver and myocardial cells are the highest, while the contents of AST and ALT in serum are very low [30]. Only when the tissue cells are damaged or the permeability is increased, will AST and ALT escape into the blood, thus increasing the activity of serum AST and ALT [31]. The activity of AST and ALT in the blood is often used as an index to diagnose the function of the heart and liver. The results showed that adding SPC at the starter stage of broilers had a tendency to decrease AST/ALT. It can be concluded that adding SPC to the starter diet of broilers can improve heart and liver function and, to a certain extent, body health. TP is a mixture of a variety of proteins, which promotes nutrition, transportation and immune function. It is conducive to maintaining the metabolism of the body and to some extent reflects the status of protein metabolism and nutrition in the body [32]. In this study, adding SPC to the starter diet of broilers has a tendency to increase TP and GLO, which suggests that SPC may improve liver function by affecting AST and ALT and thus increasing TP. An abnormal increase of T-CHO and TG content is usually a reflection of abnormal lipoprotein metabolism and it often causes kidney, cardiovascular and cerebrovascular diseases [33,34]. HDL is mainly synthesized in the liver and small intestines, which can clear cholesterol in tissues and cells, thus ensuring stable cholesterol concentrations in the body [35]. In this study, SPC12 increased the T-CHO and HDL in blood but did not affect the TG content. It can be speculated that the body of broilers may respond to elevated T-CHO by raising HDL in the blood, so as to prevent body cholesterol abnormality, remove the excess TG, maintain the health of the body and improve the production performance and health level of broilers.

Carcass yield is important for measuring the growth performance and carcass traits of animals. The results of this study showed that adding 12% SPC to the starter diet could improve the carcass yield of 42-day-old broilers but had no significant effect on other carcass traits. Similarly, Kim et al. [36] also reported that dietary supplementation with 3% FSBM products during the first seven days after hatching did not affect the relative weights of both breast muscle and abdominal fat at 35 days. Guo et al. [37] found that fermented SBM had no effect on the breast muscle index and abdominal fat yield of broilers. Moreover, partial substitution of FSBM for SBM (4.5%–6.0%) did not change the weight and relative weight of the abdominal fat in broiler chickens at 37 days [38]. The interpretation of the difference in growth performance among fermented SBMs proved difficult because all test diets were formulated to be equal in the contents of TMEn, CP and available amino acids [36]. The reason for the increase of carcass yield and growth performance were the increase of digestibility of nutrients and the decrease of ANFs in the diet. In the further study, we will design a test to verify this conclusion.

Immune organ (thymus, spleen and the bursa of fabricius) weights are indicators of the immune status of chickens and are therefore commonly used to evaluate the immunity of chickens. The thymus is a central lymphatic organ and its function is closely relevant to immunity. The thymus and bursa of fabricius are central lymphoid organs of poultry, which are essential to the ontogenetic development of adaptive immunity [39]. The thymus is the organ where T-cells develop, which provide the basis for cellular immunity [40]. Impairments of the immune system could increase susceptibility to infection, autoimmune disease and cancer and reduce growth performance, which would hinder the development of poultry industry [41,42]. The results showed that the thymus index of 42-day-old broilers was significantly increased by adding 12% SPC to the starter 1-10-day diet, which may, to a certain extent, reflect the health status of the broilers.

In this study, SPC supplementation in starter diets had no effect on meat quality. Meat quality is an important characteristic for producers and consumers, mainly described by pH, meat color and shear force indicators [43]. The pH is one of the most important parameters for meat quality, as it has a positive correlation with the water holding capacity, as well as redness and tenderness [44] and a negative correlation with the lightness [45] and drip loss of meat [46]. Mourao et al. [47] showed that muscle pH was not affected by diet, which is consistent with the results of this study. Meat color appears as an important parameter for assessing the eating quality of poultry meat [48]. In this study, adding SPC to the diet did not change the meat color of muscle, which may be caused by the same level of crude protein in the diet. 

## 5. Conclusions

The results of the study have shown that addition of SPC at the level of 12% can improve the growth performance, blood biochemical indexes, carcass traits and immune organ index of broilers and have no adverse effect on the meat quality. Nevertheless, further studies are needed to prove the mechanism of SPC supplementation in starter diets for improving broiler performance.

## Figures and Tables

**Table 1 animals-11-00281-t001:** Comparison of nutritional quality of soybean meal (SBM) and soy protein concentrate.

Item	Soy Protein Concentrate (AX3)	Soybean Meal (SBM)
Crude protein, %	68.0	43.0
Dry matter, %	92.0	88.4
Crude fat, %	2.9	2.3
Crude ash, %	3.6	7.0
Crude fiber, %	4.0	8.0
Lysine, g/kg	41.0	26.8
Methionine, g/kg	10.0	5.9
Cystine, g/kg	10.0	6.5
Threonine, g/kg	27.0	17.1
Tryptophan, g/kg	9.0	5.7
Isoleucine, g/kg	32.0	19.9
Leucine, g/kg	54.0	33.5
Valine, g/kg	34.0	20.9
Phenylalanine, g/kg	37.0	22.1
Tyrosine, g/kg	27.0	14.7
Histidine, g/kg	17.0	11.7
Arginine, g/kg	49.0	33.8
Trypsin inhibitor, mg/g	<1.5	4.2
Stachyose, %	<0.2	3.52
Raffinose, %	<0.2	2.11
Lectin, ppm	<0.1	339.4
β-conglycinin, mg/g	not detected	15.8
Phytic acid bound P %	0.3	0.39

The nutritional composition of soybean protein concentrate (AX3) and soybean meal (SBM) were determined by TripleA company.

**Table 2 animals-11-00281-t002:** Chemical composition of basal diets (as fed).

Items	Stages					
	1–10 d				11–21 d	22–42 d
	Control	SPC4	SPC8	SPC12		
Ingredients, %						
Corn	50.55	52.14	56.34	59.2	53.33	57.73
SBM (43%)	42.10	37.00	29.60	23.40	37.80	33.20
Soy oil	3.40	2.90	2.00	1.30	4.70	5.50
CaHPO_4_·2H_2_O	1.64	1.69	1.81	1.86	1.73	1.20
CaCO_3_	1.33	1.33	1.30	1.30	1.32	1.39
*DL*-Met	0.24	0.24	0.23	0.22	0.26	0.24
*L*-LysHCL (98%)	0.12	0.08	0.10	0.10	0.24	0.12
NaCl	0.30	0.30	0.30	0.30	0.30	0.30
Primix vitamin ^a^	0.03	0.03	0.03	0.03	0.03	0.03
Primix mineral ^b^	0.20	0.20	0.20	0.20	0.20	0.20
Choline chloride (70%)	0.09	0.09	0.09	0.09	0.09	0.09
AX3	0	4	8	12	0	0
Total	100	100	100	100	100	100
Nutrient level, % ^c^						
ME (kcal/kg)	2950	2950	2950	2950	3050	3150
CP	22.50	22.50	22.50	22.50	21.00	19.00
Ca	1.00	1.00	1.00	1.00	1.00	0.90
Total P	0.69	0.69	0.69	0.70	0.69	0.58
NPP	0.45	0.45	0.45	0.45	0.45	0.35
D-Lys	1.18	1.18	1.18	1.19	1.19	1.00
D-Met	0.56	0.56	0.55	0.55	0.55	0.50

^a^ Premix vitamins provided per kilogram of diet: Vitamin A (retinyl palmitate), 8000 IU; Vitamin D_3_ (cholecalciferol), 1000 IU; Vitamin E (dl-α-tocopheryl acetate), 20 IU; Vitamin K_3_ (menadione sodium bisulfate complex), 0.50 mg; Vitamin B_1_, 2.00 mg; Vitamin B_2_, 8.00 mg; Vitamin B_6_, 3.50 mg; Vitamin B_12_ (cobalamin), 0.01 mg; niacin, 35.00 mg; calcium pantothenic, 10.00 mg; folic acid, 0.55 mg; biotin, 0.18 mg. ^b^ Premix minerals provided per kilogram of diet: Fe 80.00 mg; Mn 100.00 mg; Zn 80.00 mg; I 0.70 mg; Se 0.30 mg. No Cu was supplied in mineral premix. ^c^ Nutrient level is calculated value.

**Table 3 animals-11-00281-t003:** Effects of dietary soy protein concentrate (SPC) in starter diet on growth performance of broilers.

Items	Groups				SEM	*p*-Value
	Control	SPC4	SPC8	SPC12		
1 d BW, g	45.99	45.97	46.09	46.17	0.368	0.198
1-10 d						
BW, g	321.35 ^b^	328.71 ^ab^	331.32 ^ab^	336.92 ^a^	1.957	0.029
ADG, g	27.53 ^b^	28.27 ^ab^	28.52 ^ab^	29.08 ^a^	0.194	0.029
ADFI, g	31.02 ^b^	31.76 ^ab^	32.34 ^a^	32.22 ^a^	0.169	0.015
FCR	1.13	1.11	1.13	1.11	0.006	0.419
1-21 d						
BW, g	1152.98	1173.74	1179.91	1186.47	6.841	0.336
ADG, g	52.71	53.70	53.99	54.30	0.326	0.340
ADFI, g	67.84	69.23	68.77	68.82	0.278	0.358
FCR	1.29	1.29	1.28	1.27	0.004	0.181
22-42 d						
BW, g	3135.17 ^b^	3202.72 ^ab^	3264.38 ^a^	3279.07 ^a^	19.567	0.028
ADG, g	99.11	101.12	102.64	104.63	0.850	0.115
ADFI, g	163.40	168.15	170.15	166.53	1.012	0.107
FCR	1.65	1.67	1.63	1.59	0.012	0.115
1-42 d						
ADG, g	73.55 ^b^	75.16 ^ab^	76.63 ^a^	76.98 ^a^	0.466	0.028
ADFI, g	115.62	118.69	119.46	117.67	0.563	0.086
FCR	1.57 ^a^	1.58 ^a^	1.56 ^ab^	1.53 ^b^	0.007	0.055
EPI	469.80	483.11	493.17	484.20	4.569	0.356

In a row, means assigned different lowercase letters are significantly different, *p* < 0.05. SEM = standard error of the mean. Control = SBM basal diet; SPC4 = SBM + 4% SPC; SPC8 = SBM + 8% SPC; SPC12 = SBM+12% SPC. BW = body weight; ADG = average daily gain; ADFI = average daily feed intake; FCR = feed conversion rate; EPI = European productivity index (EPI = (livability (%) × body weight (kg))/(FCR × Production Period (days)) × 100).

**Table 4 animals-11-00281-t004:** Effects of dietary SPC on blood biochemical indices of broilers at 10 d.

Items	Group				SEM	*p*-Value
	Control	SPC4	SPC8	SPC12		
ALT, U/L	12.63	13.44	14.3	15.07	0.380	0.115
AST, U/L	270.36	265.38	268.50	258.19	6.185	0.915
ALT/AST	20.76	19.82	19.31	18.07	0.360	0.055
TP, g/L	27.93	28.60	26.16	27.61	0.406	0.197
GLO, g/L	17.53	17.43	16.44	17.54	0.262	0.383
BUN, mmol/L	3.43	3.41	3.47	3.59	0.064	0.768
T-CHO, mmol/L	3.37	3.48	3.39	3.36	0.042	0.746
TG, mmol/L	0.29	0.20	0.25	0.18	0.017	0.104
HDL, mmol/L	2.18	2.12	2.09	2.06	0.035	0.664
LDL, mmol/L	0.88	0.90	0.85	0.85	0.027	0.914

In a row, means assigned different lowercase letters are significantly different, *p* < 0.05. SEM = standard error of the mean. Control = SBM basal diet; SPC4 = SBM + 4% SPC; SPC8 = SBM + 8% SPC; SPC12 = SBM + 12% SPC. ALT = Alanine aminotransferase; AST = Aspartate aminotransferase; TP = Total protein; GLO = Globulin; BUN = Urea nitrogen; T-CHO = Total cholesterol; TG = Triglycerides; HDL = High density lipoprotein; LDL = Low density lipoprotein.

**Table 5 animals-11-00281-t005:** Effects of dietary SPC on Blood Biochemical Parameters of broilers at 42 d.

Items	Group				SEM	*p*-Value
	Control	SPC4	SPC8	SPC12		
ALT, U/L	14.75	18.75	20.43	17.25	0.963	0.206
AST, U/L	426.13	426.71	514.43	432.63	19.623	0.341
ALT/AST	25.53	25.10	25.55	25.42	0.497	0.989
TP, g/L	34.01	33.84	34.10	36.90	0.501	0.080
GLO, g/L	24.20	22.79	23.09	24.86	0.335	0.080
BUN, mmol/L	3.85	4.00	3.93	4.01	0.202	0.992
T-CHO, mmol/L	2.66 ^b^	2.60 ^b^	2.61 ^b^	3.14 ^a^	0.078	0.024
TG, mmol/L	0.42	0.39	0.39	0.34	0.032	0.895
HDL, mmol/L	1.71 ^ab^	1.61 ^b^	1.60 ^b^	1.85 ^a^	0.037	0.044
LDL, mmol/L	0.76	0.74	0.86	0.91	0.051	0.592

In a row, means assigned different lowercase letters are significantly different, *p* < 0.05. SEM = standard error of the mean. Control = SBM basal diet; SPC4 = SBM+4% SPC; SPC8 = SBM+8% SPC; SPC12 = SBM + 12% SPC. ALT = Alanine aminotransferase; AST = Aspartate aminotransferase; TP = Total protein; GLO = Globulin; BUN = Urea nitrogen; T-CHO = Total cholesterol; TG = Triglycerides; HDL = High density lipoprotein; LDL = Low density lipoprotein.

**Table 6 animals-11-00281-t006:** Effects of dietary SPC in starter diet on carcass traits of broilers at 42 d.

Items	Group				SEM	*p*-Value
	Control	SPC4	SPC8	SPC12		
Carcass yield, %	92.45 ^b^	93.00 ^ab^	93.43 ^ab^	93.99 ^a^	0.215	0.043
Eviscerated with giblet yield, %	73.55	73.95	72.58	75.10	0.369	0.097
Thigh yield, %	17.73	17.09	16.67	17.02	0.211	0.367
Leg yield, %	14.12	14.14	13.92	13.79	0.122	0.729
Claw yield, %	5.17	5.25	5.27	5.22	0.058	0.949
Pectoralis major yield, %	25.60	25.74	26.13	25.93	0.310	0.942
Pectoralis minor yield, %	5.71	5.63	6.11	5.77	0.101	0.364
Wing yield, %	9.70	9.71	9.53	10.04	0.100	0.349
Abdominal fat yield, %	1.87	1.79	1.99	1.66	0.077	0.505
Heart index, g/kg	4.15	4.18	4.08	4.08	0.069	0.948
Liver index, g/kg	19.01	20.38	19.71	18.96	0.578	0.816
Pancreas index, g/kg	1.91	1.67	1.83	1.67	0.062	0.416
Gizzard index, g/kg	5.61	6.20	6.15	6.32	0.157	0.391
Glandular stomach index, g/kg	3.94	4.65	4.75	4.26	0.249	0.662

In a row, means assigned different lowercase letters are significantly different, *p* < 0.05. SEM = standard error of the mean. Control = SBM basal diet; SPC4 = SBM + 4% SPC; SPC8 = SBM + 8% SPC; SPC12 = SBM + 12% SPC.

**Table 7 animals-11-00281-t007:** Effects of dietary SPC in starter diet on immune organ index of broilers.

Items	Groups				SEM	*p*-Value
	Control	SPC4	SPC8	SPC12		
10 d						
Spleen index, g/kg	0.87	0.85	0.84	0.90	0.018	0.644
Bursa fabricius index, g/kg	2.34	2.13	1.88	2.09	0.062	0.059
42 d						
Thymus index, g/kg	1.18 ^b^	1.11 ^b^	1.20 ^b^	1.69 ^a^	0.776	0.028
Spleen index, g/kg	0.85	0.96	0.96	1.00	0.443	0.729
Bursa fabricius index, g/kg	0.68	0.63	0.71	0.58	0.434	0.740

In a row, means assigned different lowercase letters are significantly different, *p* < 0.05. SEM = standard error of the mean. Control = SBM basal diet; SPC4 = SBM + 4% SPC; SPC8 = SBM + 8% SPC; SPC12 = SBM + 12% SPC.

**Table 8 animals-11-00281-t008:** Effects of dietary SPC in starter diet on meat quality of broilers at 42 d.

Items	Groups				SEM	*p*-Value
	Control	SPC4	SPC8	SPC12		
pH	5.91	5.81	5.94	6.03	0.038	0.258
L*	55.67	57.49	57.29	55.70	0.426	0.263
a*	2.30	1.73	1.76	1.71	0.167	0.559
b*	3.43	3.29	2.85	3.49	0.226	0.766
Pressing loss, %	34.23	33.65	31.57	30.79	0.729	0.278
Shear force (kgf)	1.59	1.51	1.60	1.60	0.016	0.114

In a row, means assigned different lowercase letters are significantly different, *p* < 0.05. SEM = standard error of the mean. Control = SBM basal diet; SPC4 = SBM + 4% SPC; SPC8 = SBM + 8% SPC; SPC12 = SBM + 12% SPC. L* = lightness; a* = redness; b* = yellowness.

## Data Availability

None of the data were deposited in an official repository.

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
