# Peer review of "Effects of Soy Protein Concentrate in Starter Phase Diet on Growth Performance, Blood Biochemical Indices, Carcass Traits, Immune Organ Indices and Meat Quality of Broilers"

_animals, 2021, doi:10.3390/ani11020281_

Round 1
Reviewer 1 Report
English language: there are incomplete sentences such as in lines 20-21. The authors should read their manuscript extensively and correct mistakes.
Title: replace early diet with starter phase diet
Introduction: It is not focused. Emphasis should be laid on the effect of starter phase diet on growth performance and meat quality. Please provide more detailed information on SPC and why this product is not frequently used.
Materials and Methods: Please provide information on the manufacturer of AX3Digest.
The usual feeding diets are starter, grower and finisher. Why you name your diets prestarter, starter and grower. A pre-starter is not a diet in order to adapt to the new diet.
Lines 103 and 104: as it is written the reader understands that the birds were fed the same diet i.e. contol, SPC4, SPC8 and SPC12. In fact birds were fed the same control diet adapted to their growth stage.
Line 116: Day 42
The presentation of materials and methods and the results should follow a logical order i.e. live bird (biochemical indices), slaughtered bird (carcass characteristics), meat quality characteristics.
Line 126: Farouk et al (2004) determined WHC (water holding capacity). "Pressing loss" is a "lab slang" method description.
Line 120 broilers 32 line 127 broilers 96. Please provide detailed information on the sampling and number of samples per performed analysis.
Table 3. Please redesign the table. As it is exhaustive for the reader. Add one column with the growth day/peripd and separate data of each phase (add lines). Include mortality and provide information on EPI (what does the abbreviation EPI stands for and what is EPI)
Table 5: Shear force not share force.
The results should be presented with the following order: pH, colour, WHC and shear force.
Table 6: There is data on day 10 but there is no information in the Materials and Methods section on how this data was collected. Thymus index on day 10 is not included.
Lines 232 - 235 Please delete the sentence on fish trout. It is completely different species.
Discussion. You are speculating many of the results and there is no analysis on the levels of the antinutritional factors of the experimental diets. Please try to explain your results without speculations
Line 286-287 please delete the sentence (repetition).
Conclusions did you use AX3 as reported in Materials and Methods or AX12? Do not start the conclusions section with an abbreviation.
E.g. The results of the study have shown that addition of SPC at the level of .... can impove...
Author Response
Point 1: there are incomplete sentences such as in lines 20-21. The authors should read their manuscript extensively and correct mistakes.
Response 1: Thank you for your insightful suggestion. Revisions have been made according to this suggestion on page 1, line 20-21.
line 21-25: Since soy protein concentrate (SPC) has the characteristics of low content of antinutritional factors (ANFs), high digestibility and high amino acid content, we evaluated whether supplementation of SPC in starter diet could promote the later growth of broilers without changing the diets nutritional level (the same Metabolizable energy (ME), crude protein (CP) and amino acids (AAs)).
Point 2: Title: replace early diet with starter phase diet.
Response 2: Thank you for your insightful suggestion.
The tittle has been revised (Effects of soy protein concentrate in starter phase diet on growth performance, blood biochemical indices, carcass traits, immune organ indices and meat quality of broilers)
Point 4: Introduction: It is not focused. Emphasis should be laid on the effect of starter phase diet on growth performance and meat quality. Please provide more detailed information on SPC and why this product is not frequently used.
Response 4: Thank you for your insightful suggestion. We agree with your suggestion.
Line 75-78: On the one hand, previous studies mainly focused on adding SPC in the whole feeding stage of broilers, which led to the increase of feed cost [13]. On the other hand, the low-cost antibiotics were allowed to be added to diets to promote early growth of animals [14,15]. But in recent years, with the prohibition of antibiotics, the importance of early diet has attracted the attention of researchers.
Point 5: Materials and Methods: Please provide information on the manufacturer of AX3Digest
Response 5: Thank you for your insightful suggestion.
Line84: AX3Digest® (produced by tripleA company in Denmark) is a soy protein concentrate produced by a novel patented method.
Point 6: The usual feeding diets are starter, grower and finisher. Why you name your diets pre starter, starter and grower. A pre-starter is not a diet in order to adapt to the new diet.
Response 6: Thank you for your insightful suggestion. We agree with your suggestion.
.
Point 7: Lines 103 and 104: as it is written the reader understands that the birds were fed the same diet i.e. contol, SPC4, SPC8 and SPC12. In fact birds were fed the same control diet adapted to their growth stage
Response 7: Thank you for your insightful suggestion. I have revised this.
Lines 107 and 110: In starter phase (1-10 d), birds were fed a corn-SBM-based basal mash diet (control) and the basal diet was supplemented with SPC at 4% (SPC4), 8% (SPC8), 12% (SPC12). In the grower phase (11-21 d) and the finisher phase (22-42 d), the birds in all four treatment groups were fed the same diets.
Point 8: Line 116: Day 42
Response 5: Thank you for your insightful suggestion. We agree with your suggestion.
Line 112: The broiler chicks were individually weighed on day 1, day 11, day 22 and day 42.
Point 8: The presentation of materials and methods and the results should follow a logical order i.e. live bird (biochemical indices), slaughtered bird (carcass characteristics), meat quality characteristics
Response 8: Thank you for your insightful suggestion. We agree with your suggestion. it has been modified.
Point 9: Line 126: Farouk et al (2004) determined WHC (water holding capacity). "Pressing loss" is a "lab slang" method description
Response 9: Thank you for your insightful suggestion. We refer to the following manuscript to describe this indicator.
Huang JC, Yang J, Huang M, Chen KJ, Xu XL, Zhou GH. The effects of electrical stunning voltage on meat quality, plasma parameters, and protein solubility of broiler breast meat. Poult Sci. 2017;96(3):764-9.
Point 10: Line 120 broilers 32 line 127 broilers 96. Please provide detailed information on the sampling and number of samples per performed analysis.
Response 10: Thank you for your insightful suggestion. It has been modified
Line 150: A breast muscle sample 1.0 cm in diameter and 0.5 cm in thickness was removed from the 32 broilers for each treatment at 24 h postmortem.
Point 11: Table 3. Please redesign the table. As it is exhaustive for the reader. Add one column with the growth day/peripd and separate data of each phase (add lines). Include mortality and provide information on EPI (what does the abbreviation EPI stands for and what is EPI)
Response 11: Thank you for your insightful suggestion. I have revised this. in table 3 and line 195.
Point 12: Table 5: Shear force not share force.
Response 12: Thank you for your insightful suggestion. Table 8: Shear force
Point 13: The results should be presented with the following order: pH, colour, WHC and shear force.
Response 13: Thank you for your insightful suggestion. I have revised in table 8.
Point 14: Table 6: There is data on day 10 but there is no information in the Materials and Methods section on how this data was collected. Thymus index on day 10 is not included.
Response 14: Thank you for your insightful suggestion. I have revised this and supplemented in the materials and methods.
Line 142-144: On the day 10 and day 42, the birds were exsanguinated to obtain tissues. The spleen, thymus (on the day 10, the thymus is too small to weight, so there only index of spleen and bursa of fabricius in day 10.)
Point 15: Lines 232 - 235 Please delete the sentence on fish trout. It is completely different species.
Response 15: Thank you for your insightful suggestion. I have revised this.
Point 16: Discussion. You are speculating many of the results and there is no analysis on the levels of the antinutritional factors of the experimental diets. Please try to explain your results without speculations
Response 16: Thank you for your insightful suggestion. I have revised this. In the following experiments, we can carry out other test to explain indicators.
Point 17: Line 286-287 please delete the sentence (repetition).
Response 17: Thank you for your insightful suggestion. I have revised this.
Point 18: Conclusions did you use AX3 as reported in Materials and Methods or AX12? Do not start the conclusions section with an abbreviation.
E.g. The results of the study have shown that addition of SPC at the level of .... can impove...
Response 18: Thank you for your insightful suggestion.
line 318-321: The results of the study have shown that addition of SPC at the level of 12% can improve the growth performance, blood biochemical indexes, carcass traits and immune organ index of broilers, and have no adverse effect on the meat quality. Nevertheless, further studies are needed to prove the mechanism of SPC supplementation in pre starter diets for improving broiler performance.

Reviewer 2 Report
There are many grammatical issues with this manuscript. They include:
- Line 14 ‘Danmark’ should be written as ’Denmark’
- Line 18-19 does not make sense
- Line 21 ‘if’ should be replaced by ‘whether’
- Line 22 abbreviations are not given in full
- Line 23 ’prefer’ should be ‘improved’
- Line 24 ‘our’ should be deleted
- Line 25 ‘these’ should be ‘the’
- Line 35 needs to be clarified. At the moment it reads as though the same diets were fed through all of the feeding phases
- Lines 42 and 43 have abbreviations which are not given in full
- Line 45 – I disagree with the conclusion that SPC12 could be recommended due because in my opinion some of the immune parameters suggested negative effects on the birds
- Line 56 the phrase ‘is not perfect’ is not appropriate for a scientific publication
- Line 57 ‘their’ should be replaced by ‘the’
- Lines 62-63 the word ‘destroy’ is too strong
- Line 94 ‘The lighting schedule was continuous light for 3 days after hatch...’
- Line 99 ‘devoted’ should be ‘divided’
- Line 182 the abbreviation ‘EPI’ has not been written in full
- Line 208 the thymus gland in SPC12 birds was significantly heavier than all other groups, not just the control as stated
- Line 210 why does ‘g/kg’ appear on the right hand side of the page?
- Line 218 The T-CHO level was higher in SPC12 birds compared to all other groups, not just the controls
- Line 247 the authors suggest the experiment has obtained ‘ideal’ production performance. This is inaccurate and misleading
- Line 264 the word ‘this’ should appear between ‘However, conclusion...’
- Line 271 the word ‘rate’ should be replaced with ‘index’
- Line 276 it is not clear what study is being referred to in the increase in carcass yield
- Line 285 ‘cannot’ is the wrong word – it should be replaced by ‘did not’
- Lines 288-289 is self-evident, it does not have to be stated
- The abbreviation AX12 is used instead of SPC12 in several places after line 315
Other issues include:
- Line 77 the source (company name and location) of the AX3Digest product needs to be given
- Line 79 suggests the study compared the nutritional quality of SPC and SBM but several of the SBM parameters are sourced from other publications [13, 14]
- Line 102. I am very surprised that the control diet contains 42% SBM when the Introduction explained why this would be deleterious to the birds. So is it really an appropriate control? Is it typical of a commercial pre-starter diet?
- I am surprised that all the pre-starter diets are claimed to have the same nutritional composition (especially ME) since the oil level has been reduced by up to 70%. I do not like it when nutritional data are derived by calculation
- It appears that some birds were euthanised and dissected at day 10 but this is not stated anywhere in the Methods section. Please amend
- The equation in line 124 is specific for carcass yield, not for the other parameters as listed in lines 121-122. Indeed Table 4 shows some parameters eg liver, heart as g/kg not as %
- The equation in line 156 needs to be multiplied by 1000 to get g/kg
- I do not understand why the Discussion (lines 232-234) starts with a claim about SPC in finfish diets. It is not clear why is this relevant
- Line 243 suggests pre-starter diets are a ‘survival strategy’ but is not clear why this (if the case) would be relevant for a domesticated bird
- Line 246 why is it relevant to say how much SBM needs to be processed to make 1 tonne of SPC, especially when only 30% of the SBM was replaced with SPC and the diet with most SPC also contained 9% more corn meal (the cheaper ingredient)
- Lines 257-259 lack clarity because it mentions the both the benefits of eliminating of NSP and also the content of NSP. Which is true?
- The message being delivered in line 269 is not clear. Why would it be difficult to interpret differences in performance when all fermented SBM diets were formulated similarly? Surely it suggests that differences in performance were due to non-diet factors?
- Lines 295-297 are largely ignored when the conclusions are drawn about this experiment. The 12%SPC diet resulted in an increase in the weight of the thymus which would normally be interpreted to indicate a decrease in the health status of the birds - so why is this not risk highlighted?
- Similarly, Tables 7 and 8 seem to suggest trends of increases in ALT levels in birds that were fed on SPC. The relatively small SEM values make me wonder if the p values are reported correctly. Were they? Even if the p values are correct, doesn’t the trend suggest that the SPC increases damage to heart and liver, which contradicts the statement made in line 305?
- I disagree with the conclusions in lines 322-324 that suggest SPC improves immune organ index, blood parameters and immunity of broilers.
- There are no data provided about bird immunity so this must be removed from the manuscript
Author Response
Point 1: Line 14 ‘Danmark’ should be written as ’Denmark’
Response 1: Thank you for your insightful suggestion. We agree with your suggestion.
Line 14: TripleA a/s, Bjoernkaervej 16, DK-8783 Hornsyld, Denmark; mhm@triple-a.dk (M.H.)
Point 2: Line 18-19 does not make sense
Response 2: Thank you for your insightful suggestion. We agree with your suggestion.
Line 19-21: Therefore, in order to promote the early growth of broilers, it is necessary to feed diets with high concentration of nutrients and high digestibility in the early stage after hatching.
Point 3: Line 21 ‘if’ should be replaced by ‘whether’
Response 3: Thank you for your insightful suggestion. We agree with your suggestion.
line 23; Since soy protein concentrate (SPC) has the characteristics of low content of antinutritional factors (ANFs), high digestibility and high amino acid content, we evaluated whether supplementation of SPC in starter diet could promote the later growth of broilers without changing the diets nutritional level (the same Metabolizable energy (ME), crude protein (CP) and amino acids (AAs)).
Point 4: Line 22 abbreviations are not given in full
Response 4: Thank you for your insightful suggestion. We agree with your suggestion.
line 21-25; Since soy protein concentrate (SPC) has the characteristics of low content of antinutritional factors (ANFs), high digestibility and high amino acid content, we evaluated whether supplementation of SPC in starter diet could promote the later growth of broilers without changing the diets nutritional level (the same Metabolizable energy (ME), crude protein (CP) and amino acids (AAs)).
Point 5: Line 23 ’prefer’ should be ‘improved’
Response 5: Thank you for your insightful suggestion.
line 25; The results showed that supplementing SPC in starter diet could improve the growth performance, carcass traits and immune organ indices of broilers.
Point 6: Line 24 ‘our’ should be deleted
Response 6: Thank you for your insightful suggestion.
Line 26: The results are very promising, and SPC can be considered as a starter diet additive for commercial use.
Point 7: Line 25 ‘these’ should be ‘the’
Response 7: Thank you for your insightful suggestion.
Line 26: The results are very promising, and SPC can be considered as a starter diet additive for commercial use.
Point 8: Line 35 needs to be clarified. At the moment it reads as though the same diets were fed through all of the feeding phases
Response 8: Thank you for your insightful suggestion.
Line 34-38: The experiment was divided into three phases: in starter phase (1-10 d), birds were fed a corn-SBM-based basal mash diet (control) and the basal diet was supplemented with SPC at 4% (SPC4), 8% (SPC8), 12% (SPC12). In the grower phase (11-21 d) and the finisher phase (22-42 d), the birds in all four treatment groups were fed the same diets.
Point 9: Lines 42 and 43 have abbreviations which are not given in full
Response 9: Thank you for your insightful suggestion.
line42-line 48 Alanine aminotransferase (ALT)/ aspartate aminotransferase (AST) tended to decline in SPC12 group at 10 d (p=0.055), and total protein (TP) tended to increase in the SPC12 group at 42 d (p=0.080). The contents of total cholesterol (T-CHO) and high-density lipoprotein (HDL) were significantly elevated in the SPC12 group of broilers at 42 d (p<0.05).
Point 10: Line 45 – I disagree with the conclusion that SPC12 could be recommended due because in my opinion some of the immune parameters suggested negative effects on the birds
Response 10: Thank you for your insightful suggestion. In our results, there was no difference in spleen index and bursa index. Compared with the control group, the thymus index of SPC12 group was significantly increased, but it was within the normal range. As mentioned in the following article, the increase of thymus index is a reflection of the enhancement of immune capacity.
He, S.; Yu, Q.; He, Y.; Hu, R.; Xia, S.; He, J. Dietary resveratrol supplementation inhibits heat stress-induced high-activated innate immunity and inflammatory response in spleen of yellow-feather broilers. Poult Sci 2019, 98, 6378-6387, doi:10.3382/ps/pez471.
Point 11: Line 56 the phrase ‘is not perfect’ is not appropriate for a scientific publication Response 11: Thank you for your insightful suggestion.
Line 59: At the early stage after hatching, the development of the digestive tract is imperfect and the digestion and absorption ability are not strong
Point 12: Line 57 ‘their’ should be replaced by ‘the’
Response 12: Thank you for your insightful suggestion.
Line 60: Consequently, to promote the early growth of broilers, the diet must have a high concentration of nutrients and high digestibility
Point 13: Lines 62-63 the word ‘destroy’ is too strong
Response 13: Thank you for your insightful suggestion.
Lines 66: These antinutritional factors will damage the structure of the animal digestive tract, reduce the utilisation rate of proteins and minerals, inhibit animal feeding, disrupt the balance of the intestinal microecology, and have adverse effects on animal growth
Point 14: Line 94 ‘The lighting schedule was continuous light for 3 days after hatch...’
Response 14: Thank you for your insightful suggestion. I have revised this.
Line 101-103: The lighting schedule was continuous light for 3 days after hatch, followed by a schedule of 23 hours of light and one hour of darkness throughout the test.
Point 15: Line 99 ‘devoted’ should be ‘divided’
Response 15: Thank you for your insightful suggestion. I have revised this.
Line 105: During the whole rearing period, all broilers were randomly divided into four dietary treatments,
Point 16: Line 182 the abbreviation ‘EPI’ has not been written in full
Response 16: Thank you for your insightful suggestion.
Line 195-196: EPI = European productivity index
Point 17: Line 208 the thymus gland in SPC12 birds was significantly heavier than all other groups, not just the control as stated
Response 17: Thank you for your insightful suggestion. I have revised this.
Line 227-228: On day 42, compared to the other groups the thymus index was significantly increased (p< 0.05) in the SPC12 group.
Point 18: Line 210 why does ‘g/kg’ appear on the right hand side of the page?
Response 18: Thank you for your insightful suggestion.
Line 229-230: I have revised this.
Point 19: Line 218 The T-CHO level was higher in SPC12 birds compared to all other groups, not just the controls
Response 19: Thank you for your insightful suggestion. I have revised this.
Line 201-203: On day 42, compared to the other groups, the content of T-CHO was significantly increased (p< 0.05) in the SPC12 group.
Point 20: Line 247 the authors suggest the experiment has obtained ‘ideal’ production performance. This is inaccurate and misleading
Response 20: Thank you for your insightful suggestion. I have deleted this.
Point 21: Line 264 the word ‘this’ should appear between ‘However, conclusion...’
Response 21: Thank you for your insightful suggestion.
Line 267: However, this conclusion needs to be further verified.
Point 22: Line 271 the word ‘rate’ should be replaced with ‘index’
Response 22: Thank you for your insightful suggestion.
Line 295: Similarly, Guo et al. [34] found that fermented SBM had no effect on the breast muscle index and abdominal fat rate of broilers.
Point 23: Line 276 it is not clear what study is being referred to in the increase in carcass yield
Response 23: Thank you for your insightful suggestion.
Line 299-302: The reason for the increase of carcass yield and growth performance were the increase of digestibility of nutrients and the decrease of ANFs in the diet. In the further study, we will design a test to verify this conclusion.
Point 24: Line 285 ‘cannot’ is the wrong word – it should be replaced by ‘did not’
Response 24: Thank you for your insightful suggestion.
Line 320: In this study, adding SPC to the diet did not change the meat colour of muscle, which may be caused by the same level of crude protein in the diet.
Point 25: Lines 288-289 is self-evident, it does not have to be stated
Response 25: Thank you for your insightful suggestion. I have revised this.
Point 26: The abbreviation AX12 is used instead of SPC12 in several places after line 315
Response 26: Thank you for your insightful suggestion. I have revised this.
Lines 323-326: The results of the study have shown that addition of SPC at the level of 12% can improve the growth performance, blood biochemical indexes, carcass traits and immune organ index of broilers, and have no adverse effect on the meat quality. Nevertheless, further studies are needed to prove the mechanism of SPC supplementation in pre starter diets for improving broiler performance.
Other issues include:
Point 1: Line 77 the source (company name and location) of the AX3Digest product needs to be given
Response1: Thank you for your insightful suggestion.
Line 84: AX3Digest® (produced by tripleA company in Denmark) is a soy protein concentrate produced by a novel patented method.
Point 2: Line 79 suggests the study compared the nutritional quality of SPC and SBM but several of the SBM parameters are sourced from other publications [13, 14]
Response 2: Thank you for your insightful suggestion. In the following experiments, we can carry out other test to explain indicators.
Point 3: Line 102. I am very surprised that the control diet contains 42% SBM when the Introduction explained why this would be deleterious to the birds. So is it really an appropriate control? Is it typical of a commercial pre-starter diet?
Response 3: Thank you for your insightful suggestion. This is indeed a typical commercial of corn-SBM-based diet. Because in China, SBM is the most widely used dietary protein raw material in pigs and poultry diets, what we call harmful refers to relative to high-quality SPC. But soybean meal is a good and cheap protein raw material.
Point 4: I am surprised that all the pre-starter diets are claimed to have the same nutritional composition (especially ME) since the oil level has been reduced by up to 70%. I do not like it when nutritional data are derived by calculation
Response 4: Thank you for your insightful suggestion. The calculated value of ME is generally used. The value of our raw material is obtained from the Chinese raw material database for poultry. The deviation from the measured value should be small, and it should be credible. We can see in many published articles that they also use the calculated value.
Point 5: It appears that some birds were euthanised and dissected at day 10 but this is not stated anywhere in the Methods section. Please amend
Response 5: Thank you for your insightful suggestion.
line 121: After cervical dislocation, blood samples (3 mL) were collected from the jugular vein of eight birds in each group, at the following time points: 10 and 42 days of life.
Point 6: The equation in line 124 is specific for carcass yield, not for the other parameters as listed in lines 121-122. Indeed Table 4 shows some parameters eg liver, heart as g/kg not as %
Response 6: Thank you for your insightful suggestion.
line 140: organ indices = organ weight / live body weight×1000
Point 7: The equation in line 156 needs to be multiplied by 1000 to get g/kg
Response 7: Thank you for your insightful suggestion.
line 147: Immune organ indices = Immune organ weight / live body weight×1000
Point 8: I do not understand why the Discussion (lines 232-234) starts with a claim about SPC in finfish diets. It is not clear why is this relevant
Response 8: Thank you for your insightful suggestion. I have revised this.
Point 9: Line 243 suggests pre-starter diets are a ‘survival strategy’ but is not clear why this (if the case) would be relevant for a domesticated bird
Response 9: Thank you for your insightful suggestion. Improving the quality of nutrients in the early diet of birds has a good effect on the growth and development of the later phrase, which is equivalent to reducing the economic cost and increasing the economic benefit. In the past, antibiotics were not banned in China. Antibiotics can be added to feed to promote the early growth and development of broilers. In recent years, antibiotics have been banned in China. Therefore, a new high utilization ration diet is needed to promote the early growth and development of broilers, which can also be considered as a survival strategy
Point 10: Line 246 why is it relevant to say how much SBM needs to be processed to make 1 tonne of SPC, especially when only 30% of the SBM was replaced with SPC and the diet with most SPC also contained 9% more corn meal (the cheaper ingredient)
Response 10: Thank you for your insightful suggestion. I have deleted this.
Point 11: Lines 257-259 lack clarity because it mentions the both the benefits of eliminating of NSP and also the content of NSP. Which is true?
Response 11: P Thank you for your insightful suggestion.
Line 260-262: These increases may be due to the elimination of most nonstarch polysaccharides (NSP) in SPC, which improves the early nutritional absorption capacity of broilers
Point 12: The message being delivered in line 269 is not clear. Why would it be difficult to interpret differences in performance when all fermented SBM diets were formulated similarly? Surely it suggests that differences in performance were due to non-diet factors?
Response 12: Thank you for your insightful suggestion. The reason for the increase of carcass yield and growth performance were the increase of digestibility of nutrients and the decrease of ANFs in the diet. In the further study, we will design a test to verify this conclusion.
Point 13: Lines 295-297 are largely ignored when the conclusions are drawn about this experiment. The 12%SPC diet resulted in an increase in the weight of the thymus which would normally be interpreted to indicate a decrease in the health status of the birds - so why is this not risk highlighted?
Response 13: Thank you for your insightful suggestion. In our results, there was no difference in spleen index and bursa index. Compared with the control group, the thymus index of SPC12 group was significantly increased, but it was within the normal range. As mentioned in the following article, the increase of thymus index is a reflection of the enhancement of immune capacity.
He, S.; Yu, Q.; He, Y.; Hu, R.; Xia, S.; He, J. Dietary resveratrol supplementation inhibits heat stress-induced high-activated innate immunity and inflammatory response in spleen of yellow-feather broilers. Poult Sci 2019, 98, 6378-6387, doi:10.3382/ps/pez471.
Point 15: Similarly, Tables 7 and 8 seem to suggest trends of increases in ALT levels in birds that were fed on SPC. The relatively small SEM values make me wonder if the p values are reported correctly. Were they? Even if the p values are correct, doesn’t the trend suggest that the SPC increases damage to heart and liver, which contradicts the statement made in line 305?
Response 15: Thank you for your insightful suggestion. First of all, I guarantee that the data is reliable, and the p value is analysed according to the one-way anova in spss22. Secondly, although ALT has a rising trend in SPC diet, it has not reached a significant level, and the proportion of alt/ast in spc12 diet is decreased.
Point 16: disagree with the conclusions in lines 322-324 that suggest SPC improves immune organ index, blood parameters and immunity of broilers. There are no data provided about bird immunity so this must be removed from the manuscript
Response 16: Thank you for your insightful suggestion. I have revised this.
Line 323-326: The results of the study have shown that addition of SPC at the level of 12% can improve the growth performance, blood biochemical indexes, carcass traits and immune organ index of broilers, and have no adverse effect on the meat quality. Nevertheless, further studies are needed to prove the mechanism of SPC supplementation in pre starter diets for improving broiler performance.

Reviewer 3 Report
The Authors have investigated an interesting topic and the theme has been properly described.
I would like to congratulate Authors for the good-quality of the article, the literature reported used to write the paper, and for the clear and appropriate structure. The manuscript is well written, presented and discussed, and understandable to a specialist readership.
In general, the organization and the structure of the article are satisfactory and in agreement with the journal instructions for authors. The subject is adequate with the overall journal scope.
The work shows a conscientious study in which a very exhaustive discussion of the literature available has been carried out. The introduction provides sufficient background, and the other sections include results clearly presented and analyzed exhaustively.
So, I recommend the acceptance of the paper in Animals.
Author Response
Thank you very much for your approval of our manuscript
Reviewer 4 Report
In the evaluated study it was assumed that early feeding of chickens, limited to the first 10 days of live (starter phase), determines the physiology and growth performance effects in the later period. They also assumed that replacing a port of the soybean meal with a soybean protein concentrate would reduce antinutritional factors (ANFs) in diets and would improve growth performance, selected blood biochemical indices and immune organ index, as well as slaughter yield and meat quality.
The weakness of the study is the lack of own analyzes of the chemical composition of diets, first of all the content of ANFs (table 1 gives data from the literature, and table 2 gives the calculated nutritional value of diets, without data on ANFs) and little specific research methods, not relating to the possible effects of soybean ANFs. For this reason, I estimate the scientific value of the presented results as moderate, but I do not exclude its publication (I leave the decision to the managing editor).
Author Response
Thank you very much for your comments on our manuscript.
Round 2
Reviewer 1 Report
English language corrections are needed.
Please write the aim of the study (lines 79-81). The aim of the study was....
Table 1. Please explain how soya protein (meal and concentrate) composition was determined. Do not use references in the table. If you have the analysis from a company or a previous study please state it.
Pressing loss: Please add the correct reference
"Huang JC, Yang J, Huang M, Chen KJ, Xu XL, Zhou GH. The effects of electrical stunning voltage on meat quality, plasma parameters, and protein solubility of broiler breast meat. Poult Sci. 2017;96(3):764-9." in the reference list [18]. In any case pressing loss is water holding capacity and this is the term you are using in your discussion. I find it confusing and this term is not as clear as WHC.
Table 3. Please explain how EPI is calculated and what is the purpose of this index. The index is not discussed in the Discussion section
Please check your text and replace pre-starter with starter i.e. line 326 you write pre-starter.
Dear Editors,
Thank you very much for your hard and kind work on our paper (Manuscript ID: animals-982955). We feel lucky that our manuscript went to these reviewers as the -valuable comments from them not only helped us with the improvement of our manuscript, but suggested some neat ideas for further studies. We have studied comments carefully and have made correction that we hope meet with approval.
Please find below details regarding all the changes we have made to the manuscript in response to the editor’s and reviewers’ comments. All the changes are marked in red in both this response letter and the revised manuscript.
Reply to Academic Editor
Point 1: English language corrections are needed.
Response 1: Thank you for your insightful suggestion. We have revised this.
Point 2: Please write the aim of the study (lines 79-81). The aim of the study was....
Response 2: Thank you for your insightful suggestion. We agree with your suggestion.
Line 79-81: The aim of the study was added SPC to the starter diets of broilers to determine its effect on growth performance, carcass traits, meat quality, immune organ indices and the blood biochemical indices of broilers, in order to provide evidence for the practical production of broilers.
Point 3: Table 1. Please explain how soya protein (meal and concentrate) composition was determined. Do not use references in the table. If you have the analysis from a company or a previous study please state it.
Response 3: Thank you for your insightful suggestion. We agree with your suggestion.
The nutritional composition of soybean protein concentrate (AX3) and soybean meal (SBM) were determined by TripleA company.
Point 4: Pressing loss: Please add the correct reference
"Huang JC, Yang J, Huang M, Chen KJ, Xu XL, Zhou GH. The effects of electrical stunning voltage on meat quality, plasma parameters, and protein solubility of broiler breast meat. Poult Sci. 2017;96(3):764-9." in the reference list [18]. In any case pressing loss is water holding capacity and this is the term you are using in your discussion. I find it confusing and this term is not as clear as WHC.
Response 4: Thank you for your insightful suggestion. Because the water holding capacity and pressing loss are two different concepts, in our article, we measure the pressing loss.
Point 5: Table 3. Please explain how EPI is calculated and what is the purpose of this index. The index is not discussed in the Discussion section
Response 5: Thank you for your insightful suggestion. We agree with your suggestion.
EPI = European productivity index (EPI = (liveability (%) × body weight (kg)) / (FCR × Production Period (days)) × 100).
Point 6: Please check your text and replace pre-starter with starter i.e. line 326 you write pre-starter.
Response 6: Thank you for your insightful suggestion. We have revised this in line 326
Thanks for your patience in reading this long letter. We hope that our responses and revisions are acceptable. Please do not hesitate to contact me if more revisions are required.
Yours sincerely,